# Circumstances of human conflicts with bears and patterns of bear maul injuries in Bhutan: Review of records 2015–2019

**Dorji Penjor**[1]☯, **Thinley Dorji**[2]☯¤*

1 Department of Ear, Nose and Throat Surgery, Jigme Dorji Wangchuck National Referral Hospital, Thimphu, Bhutan, 2 Kidu Mobile Medical Unit, His Majesty's People's Project, Thimphu, Bhutan

☯ These authors contributed equally to this work.
¤ Current address: Department of Internal Medicine, Armed Forces Medical College, Maharashtra University of Health Sciences, Pune, India
* dorji.thinleydr@gmail.com

**Data Availability Statement:** All relevant data are in the paper and its Supporting Information files.

**Funding:** The author(s) received no specific funding for this work.

## Abstract

Bhutan is one of the biological hotspots in the world where humans and natural flora and fauna co-exist in close proximity. Bhutan is home to two species of bears: Sloth Bear and Himalayan Black Bear. Human conflicts with bears are reported from all over the country. This study describes the profile of the victims and the pattern of injury resulting from bear attacks and circumstances around human conflicts with bears in Bhutan between 2015 and 2019. This was a cross-sectional study with a review of hospital records of patients treated at the National Referral Hospital from 01 January 2015 till 31 December 2019. Data were extracted into a structured pro forma and entered into EpiData Entry 3.1 and analysed in STATA 13.1. There were thirty-four patients who were provided care for bear maul injuries, with an average annual caseload of 6.8 cases per year. The injury prevalence was 100% and the kill prevalence was 0%. Bear attacks were reported from fourteen of twenty districts of the country. The mean age of the victims was 49 (±13) years. Males (26, 76%) and farmers (26, 76%) were the common victims; the risk of bear attacks was 0.16 per 100,000 farmers per year. The commonest region of the body attacked was the face (29, 85%) and victims were provided emergency and rehabilitative care within and outside the country. Thirty-three victims (97%) were provided post-exposure prophylaxis for rabies. All victims received antibiotics despite the lack of national guidelines on the choice of antibiotics post-bear maul. Human-bear conflict is multi-faceted, puts a considerable strain on bear-conservation efforts and requires multi-disciplinary efforts in the prevention of human injury and socioeconomic losses.

## Introduction

Bhutan is one of the biological hotspots in the world where humans and natural flora and fauna co-exist in close proximity. Located in the eastern Himalayas, 80.9% of Bhutan's total land area of approximately 38,394 square kilometres is under forest cover and more than half

**Competing interests:** The authors have declared that no competing interests exist.

of the country's landscape (51.4%) is under government-protected areas [1,2]. The relief of Bhutan ranges from 160 metres to more than 7,000 metres above sea level (masl), and allows for three eco-floristic zones: sub-tropical zone between 150 to 2000 masl, temperate zone between 2000 to 4000 masl and the alpine zone above 4000 masl [1–3]. Bhutan has a population of 681,720 persons that is growing at a rate of 1.3% per annum [4]. Population growth and socio-economic development activities have led to the increased rates of conversion of arable land and forests into other land uses such as farms, farm roads, electricity transmission lines, industries, and human settlements [2] The rural farming community comprise of 62.2% of the population and live in close proximity to natural and community forests. Human-wildlife conflicts with elephants, wild boars, deer, monkeys and bears are not uncommon across the Himalayas but lack of reporting has made it seem a rare phenomenon [5,6].

Bears are mammals that are found in the temperate climatic zones in Bhutan [3]. Bhutan is home to the Sloth Bear (*Melursus ursinus*) and the Himalayan Black Bear (*Ursus thibetanus laniger*), both categorised as vulnerable species by the International Union for Conservation of Nature Red List [3,7]. The Himalayan Black Bear, under Schedule I of the Forest and Nature Conservation Act of Bhutan 1995, is declared totally protected whether or not in government-protected areas [8]. Many of the human settlements in Bhutan are found in this belt including traditional migratory herders that move around with their yaks and sheep [9].

Bear attacks in Bhutan resulting in grievous injuries to humans are reported from several districts in national newspaper [5] Bears are strong and agile animals that defend themselves, their young ones and their territory if they feel threatened [10]. Bears have evolved a denning behaviour that minimizes metabolism and energy to survive harsh environmental conditions, primarily a seasonal lack of food and unfavourable weather [11,12]. The den entry timing of the year is influenced by many biotic and abiotic factors and more bear attacks are known to occur in the pre-denning and the denning period when bears forage for food [12]. Encounters with bears can be "sudden" where neither the person nor the bear is aware of each other's presence (surprise encounter), "provoked" when humans trespass into bear's territory (harassed bear), and "predatory" when the bear treats its victim as food [10,13]. Human conflicts with bears also pose a threat to the conservation efforts of the animal [9,14].

In Bhutan, the victims of bear maul injuries are mostly referred to the Jigme Dorji Wangchuck National Referral (JDWNR) Hospital, Thimphu where surgical and intensive care facilities are available. In this paper, we describe the socio-demographic profile of victims, the pattern of bear maul injury, and circumstances of bear attacks among those patients treated at the JDWNR Hospital, Thimphu between 01 January 2015 and 31 December 2019. A better understanding of bear attacks may help in formulating specific measures to prevent human-bear conflicts.

## Materials and method

### Study design

This was a cross-sectional study with a review of hospital records.

### Setting

Healthcare in Bhutan is provided through a three-tiered system: basic health units (recently renamed as primary health centre) and outreach clinics at the primary level, district and general hospitals at the secondary level, and referral hospitals with specialist services at the tertiary level [15]. The three referral centres are located in geographically strategic locations in the west (Thimphu), east (Monggar) and central (Gelegphu) regions.

For acute trauma cases resulting from bear attacks, the first point of contact for victims is the primary health centre that covers ninety-five percent of the population within three walking hours [1]. Health assistants or general duty medical officers are available to provide basic resuscitation. The patients are then referred to the nearest higher centre, district or general hospitals, where general doctors are available for basic surgical and wound management. Cases requiring specialist surgical, Ear, Nose and Throat (ENT), oro-maxillofacial, orthopaedic or intensive care are referred to the referral hospitals or the JDWNR Hospital by road or air [15–17]. For those cases that require further management such as plastic and reconstructive surgery, patients are referred to hospitals outside the country. The cost of referral is borne by the Royal Government of Bhutan [15].

## Study site

This study was conducted at the JDWNR Hospital, Thimphu, Bhutan.

## Study population, sample size and study period

We studied the hospital records of all patients who presented or were referred to the JDWNR Hospital, Thimphu for treatment and care related to bear maul or bear attacks between 01 January 2015 and 31 December 2019. The physical records of 34 bear maul victims were retrieved between October-December 2019.

## Data variables, sources and data collection

Patient files were extracted from the hospital's Medical Records Section. Search words such as bear maul and bear attack, and ICD-10 coding [18] of W55.89 for animal attacks related to being run over, stepped on and struck by animals were used to identify the patient records from the electronic database. The patient admission registers at the ENT Ward, Surgical and Orthopaedic Surgery Wards were physically searched for patient's name and identity for those admitted with bear attacks. All variables were extracted into a structured pro forma. Duplication of patients from readmission for repeat surgeries or treatment was avoided through careful evaluation of the patient's name and hospital registration number.

## Data analysis and statistics

Data were double entered in January 2020, validated and analysed using EpiData (version 3.1, EpiData Association, Odense, Denmark) and analysed in STATA (version 13.1, StataCorp LP USA).

The injury prevalence and kill prevalence were calculated as the percentage of the total number of patients suffering only injury and the number of patients who died of the bear attack respectively. The burden of bear attacks per year is presented as frequencies. The temporal pattern is described in terms of hours of the day and four seasons of the year (winter: December-February; spring: March-May; summer: June-August; autumn: September-November). The place and circumstance of the attack, socio-demographic profile of the patient, pattern of injury and the complications of the attack are described as frequency and proportions for categorical variables and mean (standard deviation) for continuous variables.

## Ethics approval

Ethics approval was granted by the Research Ethics Board of Health, Ministry of Health, Bhutan (REBH/Approval/2019/025 dated 12/06/2019). Informed consent from patients was

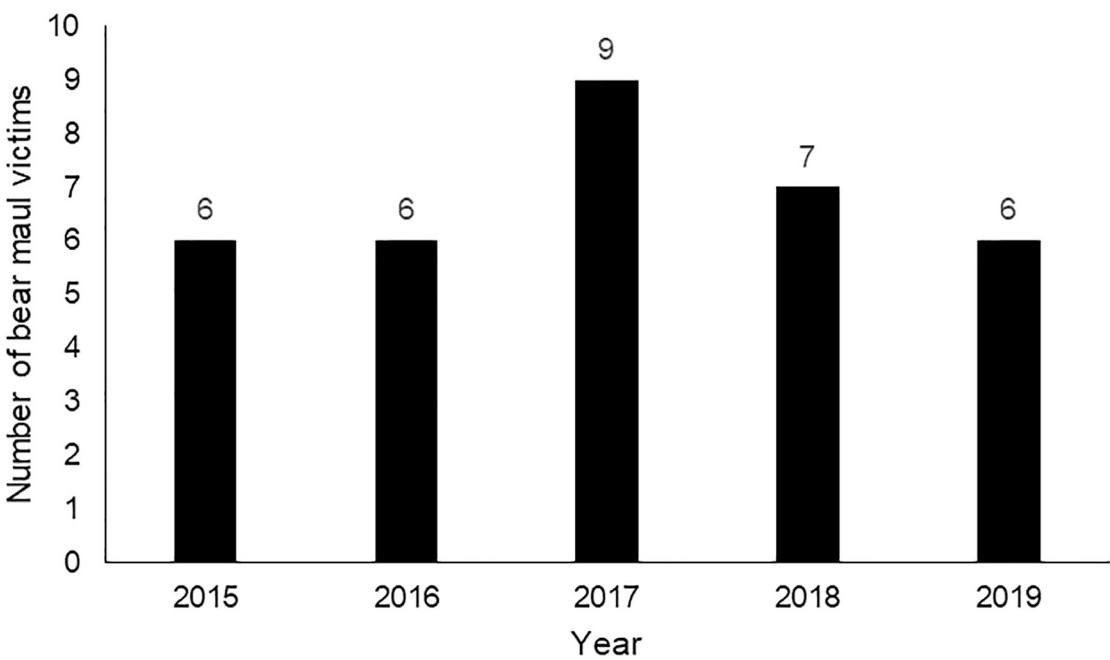

**Fig 1. The distribution of victims of bear attacks by years who received care at the Jigme Dorji Wangchuck National Referral Hospital, Bhutan during the period 2015–2019.**

waived off by the ethics review board as only de-identified data were collected. Permission for the use of hospital records was obtained from the hospital administrators.

## Results

Thirty-four patients were provided care for bear maul injuries, with an average annual caseload of 6.8 cases per year (Fig 1). The mean age of the victims was 49 (±13) years and the profile of the victims is shown in Table 1. The injury prevalence was 100% and the kill prevalence was 0%. The description of the circumstance of bear attacks is shown in Table 2; the year-wise caseload is given in Fig 1 and the incident hotspot by district is given in Fig 2.

The most common region of the body attacked was the face (29, 85%) and the details of injury assessment are shown in Table 3. The mean duration of stay in hospital in their first admission was 15 (±12) days while only one patient was treated as out-patient. Four patients (12%) required intensive care unit admission for a mean duration of 3 (±2) days. The treatment provided to the victims and the outcomes are shown in Table 4.

## Discussion

### Circumstance of bear maul incidents

The average annual bear attack rate from 2015 to 2019 was 6.8 cases per year, and attacks were reported from fourteen of twenty districts of the country. Farmers constituted the majority of victims. With the annual attack rate and the population of farmers in the country, the risk of bear attacks is 0.16 per 100,000 farmers per year. Bear attacks were closely related to occupation as the majority of the victims were farmers. The attacks were highest in September to November and were associated with pre-denning and den entry timings of bears [11,12]. While data from Bhutan are not available, data from Japan suggests that the Asiatic bear enters

**Table 1. The socio-demographic profile of victims of bear attacks who received care at the Jigme Dorji Wang-chuck National Referral Hospital, Bhutan from 2015 to 2019.**

| Bear maul victim profile | n | (%) |
|---|---|---|
| **Total** | **34** | **(100)** |
| **Age group** | | |
| 21–30 years | 4 | (12) |
| 31–40 years | 5 | (15) |
| 41–50 years | 9 | (26) |
| 51–60 years | 8 | (24) |
| $\geq$ 61 years | 8 | (24) |
| **Sex** | | |
| Male | 26 | (76) |
| Female | 8 | (24) |
| **Occupation** | | |
| Farmer | 26 | (76) |
| Cow herder | 3 | (9) |
| Soldier | 2 | (6) |
| Forest officer | 1 | (3) |
| Others | 2 | (6) |
| **First health centre of presentation***  | | |
| Basic Health Unit Grade 2 | 3 | (9) |
| Basic Health Unit Grade 1 | 2 | (6) |
| District or General Hospitals | 22 | (65) |
| Regional Referral Hospitals | 2 | (6) |
| National Referral Hospital | 5 | (15) |
| **Mode of evacuation to National Hospital** | | |
| Vehicle ambulance | 31 | (91) |
| Air ambulance | 3 | (9) |

*From January 2020, Basic Health Unit Grade 2 are renamed as Primary Health Centre, and Basic Health Unit Grade 1 are renamed as 10-bedded Hospital.

its den in November and December [11]. The second-highest number of attacks occurred between June to August. This is the period when farmers enter the forested areas to collect both timber and non-timber forest products and grazing of their cattle for economic livelihood [6,12]. Hunting is not a cause of any of the bear encounters in this study as hunting and killing of animals is strongly discouraged under the religious ethos in Bhutan and is prohibited under Forest and Nature Conservation Act of Bhutan 1995 [8]. The majority of the incidents have occurred during day time as reported from other studies in Scandinavia and India [19,20]. Males (26, 76%) were the common victims as they are more involved in outdoor activities such as hunting in Scandinavia or collection of forest products in the Himalayas and India [10,17,19–21] In Bhutan, the highest number was reported from Zhemgang district (7, 21%), one of the districts heavily dependent on agriculture and forests for livelihood [4].

Bear attacks were reported from farmlands and settlements; bear sightings in communities and urban suburbs have been reported in a community survey in Wangchuck Centennial National Park and in national media [9]. With increasing urbanization and conversion of forested areas into human settlements and roadways, bears enter their dens earlier. This may be due to increased access to food sources near roads, which allow them to gain enough fat reserves to den early [12]. This behavioural change also results in damage to cash crops such as

**Table 2. The description of the incidents of bear attacks among patients who received care at the Jigme Dorji Wangchuck National Referral Hospital, Bhutan from 2015 to 2019.**

| Circumstance of bear attack | n | (%) |
|---|---|---|
| **Season of attack** | | |
| Winter (December-February) | 6 | (18) |
| Spring (March-May) | 4 | (12) |
| Summer (June-August) | 11 | (32) |
| Autumn (September-November) | 13 | (38) |
| **Time of attack[1]** | | |
| Morning | 5 | (17) |
| Day time | 21 | (70) |
| Evening | 4 | (13) |
| Night | 0 | (0) |
| **Place of bear attack[*2]** | | |
| Forests | 11 | (69) |
| Farmlands | 2 | (13) |
| Settlements | 3 | (19) |
| **Type of place[3]** | | |
| Rural | 24 | (100) |
| Urban | 0 | (0) |
| **Type of attack[4]** | | |
| Sudden | 16 | (100) |
| Provoked | 0 | (0) |
| Predatory | 0 | (0) |
| **Number of bears that attacked[4]** | | |
| One | 16 | (100) |
| **Bear with her cub** | 3 | (9) |
| **Victim's first response[5]** | | |
| Run away | 4 | (31) |
| Fight | 8 | (62) |
| Drop dead | 1 | (8) |
| **Presence of human witness to the incident** | 5 | (15) |

*Forests* are thick jungles several hours away from the nearest human settlement. *Human settlements* are villages and towns where humans have built houses for settlements. Orchards and gardens around the human houses are considered as human settlements. *Farmlands* are considered those farming areas that are away from human settlements where farmers or villagers may have to travel some distance to get to the farmland.

[1]Missing = 4;

[2]Missing = 18;

[3]Missing = 10;

[4]Missing = 18;

[5]Missing = 21.

fruits and corns and livestock during the predenning period as reported from Wangchuck Centennial National Park, Bhutan [6,9,12].

In our study, the majority of the incidents were sudden attacks and the majority fought back with the bear. The common weapons used for fighting back were sticks, daggers and *patangs* (heavy metallic sword) and no firearms were reported in this study. Various other responses such as running away and dropping dead have been reported. Presently, there is not

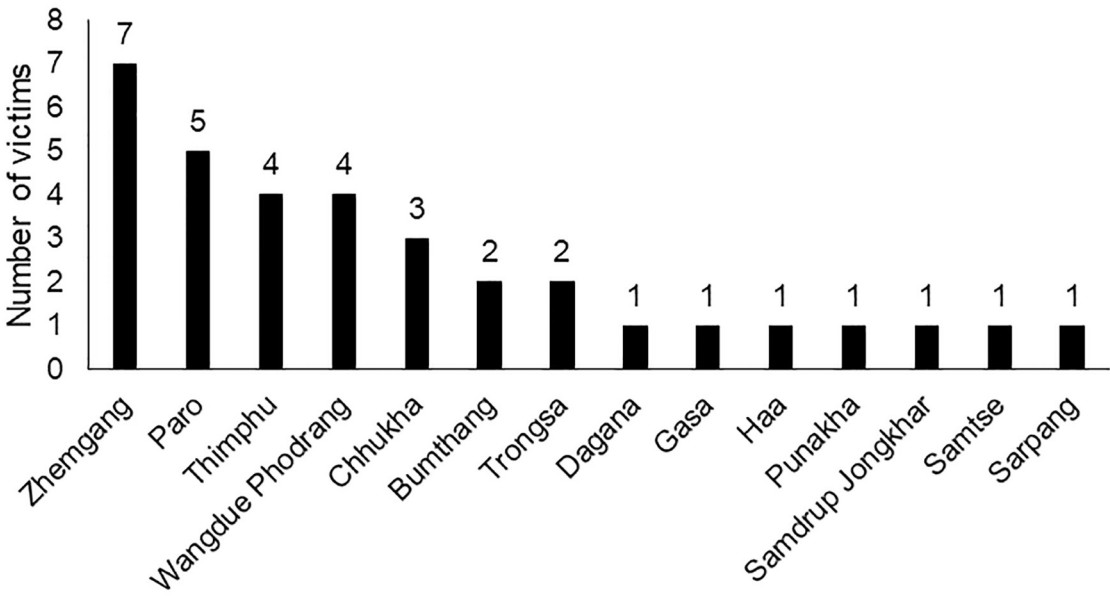

**Fig 2. The distribution of victims of bear attacks by districts who received care at the Jigme Dorji Wangchuck National Referral Hospital, Bhutan during the period 2015–2019.**

enough evidence on whether fighting, running away or dropping dead results in lesser injuries or fatalities in bear attacks with *Melursus ursinus* and *Ursus thibetanus laniger*.

## Bear maul injuries and clinical management

All bear maul victims sustained injuries and there were no deaths. Bears are considered intelligent animals that target the upper body to weaken the enemy and prevent retaliation [6]. The face, head and upper limbs were the common areas where injuries were sustained. Facial injuries include not only soft tissue and nerve injuries but also fractures of the facial and skull bones. Evaluation and management require a specialist evaluation and in Bhutan, such facilities are available only in the regional or national referral hospitals.

The geographic terrain in the Himalayas is a major challenge in providing emergency resuscitation and surgical care of victims of bear attacks. While transportation and road networks are progressively improving in Bhutan, the emergency air medical service, Bhutan Aeromedical Retrieval (BEAR), catered to three airlifts of victims to the JDWNR Hospital, Thimphu in its first year of service from 2017 to 2018 [22].

The majority of the victims were treated within Bhutan with wound debridement, skin graft and tissue flaps. While bear maul incidents were reported from 14 districts, access to surgical treatment is available only at three tertiary hospitals. Seven patients were referred outside the country for plastic and reconstructive surgery, the cost of which is borne by the Royal Government of Bhutan [15]. This reflects the need to expand surgical services and increase the capacity of specialist surgical services in the country.

The pattern of injuries can predict the functional and economic losses. Attacks were sustained on upper limbs resulting in loss of tissue, nerve damage and amputations resulting in loss of socioeconomic function. The entire cost for healthcare and treatment of the victim is

**Table 3. The description of the injury sustained by the victims of bear attacks who received care at the Jigme Dorji Wangchuck National Referral Hospital, Bhutan from 2015 to 2019.**

| Injury assessment[*] | n | (%) |
|---|---|---|
| **Body part injured** | | |
| Face | 29 | (85) |
| Head | 23 | (68) |
| Neck | 2 | (6) |
| Chest | 3 | (9) |
| Back | 4 | (12) |
| Abdomen | 1 | (3) |
| Upper limbs | 23 | (68) |
| Lower limbs | 5 | (15) |
| Genital area | 0 | (0) |
| **Regions of the face injured** | | |
| Forehead | 20 | (59) |
| Eyes | 17 | (50) |
| Ears | 7 | (21) |
| Nose | 15 | (44) |
| Lips | 9 | (27) |
| Cheeks | 15 | (44) |
| Chin | 2 | (6) |
| **Tissues involved in injury** | | |
| Soft tissue | 34 | (100) |
| Bones (fracture) | 23 | (68) |
| Viscera | 0 | (0) |
| **Type of injury** | | |
| Deep laceration | 33 | (97) |
| Loss of tissue | 10 | (29) |
| Puncture wounds | 18 | (52) |
| Bone fracture (any bone) | 23 | (68) |
| Damage of nerves | 1 | (3) |
| **Bones fractured: *Facial bones*** | | |
| Zygoma | 4 | (17) |
| Nasal bones | 11 | (48) |
| Maxilla | 9 | (40) |
| Mandible | 3 | (13) |
| Orbital walls | 5 | (22) |
| Nasoethmoid areas | 3 | (13) |
| **Bones fractured: *Other bones*** | | |
| Phalanges | 1 | (4) |
| Metacarpals | 4 | (17) |
| Ulna | 2 | (9) |
| Skull bone | 1 | (4) |
| **Type of bone fractures** | | |
| Simple | 7 | (30) |
| Compound | 12 | (52) |
| Comminuted | 14 | (61) |

[*]Multiple responses allowed.

**Table 4. Treatments offered to the victims of bear attacks at the Jigme Dorji Wangchuck National Referral Hospital, Bhutan from 2015 to 2019.**

| Treatment provided | n | (%) |
|---|---|---|
| **Tetanus toxoid**[1] | 30 | (100) |
| **Rabies post-exposure prophylaxis**[2] | | |
| Anti-rabies vaccine | 19 | (83) |
| Rabies immunoglobulin | 3 | (13) |
| Both | 3 | (13) |
| None | 1 | (4) |
| **Antibiotics** | 34 | (100) |
| Metronidazole | 28 | (82) |
| Ampicillin | 22 | (65) |
| Cefazolin | 14 | (41) |
| Gentamicin | 14 | (41) |
| Ceftriaxone | 13 | (38) |
| Amoxicillin | 1 | (3) |
| Ciprofloxacin | 1 | (3) |
| **Wound debridement** | 34 | (100) |
| **Skin graft** | 4 | (12) |
| **Flaps** | 4 | (12) |
| **Tracheostomy** | 1 | (3) |
| **Intensive care required** | 4 | (12) |
| **Closure of wound** | | |
| Primary | 27 | (79) |
| Secondary | 7 | (21) |
| **Treatment outcome** | | |
| Discharged | 27 | (79) |
| Referred outside the country* | 7 | (21) |
| Died | 0 | (0) |

*Referral outside the country to centres in India.

[1]Missing = 4;

[2]Missing = 11.

borne by the Royal Government and compensation for cash crop damage is also given by the Royal Government. There is no compensation for the injury and loss of function of the victim unlike in Nepal or in selected states in India [6,20].

All patients in our study had received anti-rabies vaccination and/or rabies immunoglobulin. Though 59% of 59,000 annual global deaths related to human rabies occur in Asia, the majority are transmitted by rabid dogs [23]. While rabies infection is documented among bears in North America, the seroprevalence of rabies antibody among sixty-three black bears in North America was 5.5% [24] and there is a lack of literature on human transmission of rabies from a rabid bear. However, the majority of rabies guidelines across Asia including the National Guideline for Management of Rabies 2014, Bhutan recommend post-exposure prophylaxis with anti-rabies vaccine and rabies immunoglobulin [23,25].

In our study, all the patients received antibiotics. Antibiotic administration is guided by case reports and bacteriology of wound infection post bear maul injuries [26,27] Based on available case reports, cultures from actual bear bite infection that isolated polymicrobial growth, antimicrobial coverage recommended are Gram positives with either penicillin or

first-generation cephalosporin and include broader coverage for Gram-negative organisms [26,27].

**Bear conservation efforts.** Apart from the human injuries and socioeconomic loss resulting from bear maul incidents, human-bear conflict is multi-faceted and puts a considerable strain on bear-conservation efforts. With frequent conflicts with bear resulting in the depredation of livestock and crops and human injury that are reported in national media [28], people in living in bear conflict areas see them as pest to their livelihood and favour culling Asiatic black bears [9]. Such frustrations and retaliations are reported as serious limiting factors for bear conservation in Bhutan and in other countries [9,29].

The Royal Government has initiated the following community and household-based protection measures to reduce human-wildlife conflicts: electric fences to prevent wildlife movement towards human settlements, building predator-proof corrals to prevent livestock loss by predators at night, and the planting of crops that are unpalatable to wildlife, such as peppermint. The government has also initiated educational campaigns aimed at farmers on how to avoid bear encounters and first aid methods in an event of bear maul injury. However, the effects of education on reduction of bear attacks have not yet been demonstrated from experiences reported elsewhere [19]. Therefore, there is a need to identify and implement measures with the engagement of local communities to conserve natural bear habitats to allow diversionary feeding and reduce anthropogenic food sources for bears [9,29].

## Limitations

This study reviewed only those patients who survived to present to the JDWNR Hospital. Those with minor injuries may be treated in other hospitals. However, this number is expected to be minimum as the majority are usually referred to the JDWNR Hospital. This paper focussed on the pattern of injury in the immediate post-injury period. The psychological impact of disfigurement and disability resulting from such injuries were not studied.

## Conclusions

The average annual bear attack rate was 6.8 cases per year and attacks were reported from fourteen of twenty districts of the country. The injury prevalence was 100% with the majority in the face and upper limbs. The circumstances of human-bear conflict and patterns of injuries should be understood in the context of bear conservation efforts in the country.

## Supporting information

**S1 Data.**
(DTA)

## Acknowledgments

The authors are grateful to the hospital administration and the staff who helped in manual extraction of patient files.

## Author Contributions

**Conceptualization:** Dorji Penjor, Thinley Dorji.

**Data curation:** Dorji Penjor, Thinley Dorji.

**Formal analysis:** Dorji Penjor, Thinley Dorji.

**Investigation:** Dorji Penjor, Thinley Dorji.

**Methodology:** Dorji Penjor, Thinley Dorji.

**Project administration:** Dorji Penjor.

**Resources:** Dorji Penjor, Thinley Dorji.

**Software:** Dorji Penjor, Thinley Dorji.

**Supervision:** Dorji Penjor, Thinley Dorji.

**Validation:** Dorji Penjor, Thinley Dorji.

**Visualization:** Dorji Penjor, Thinley Dorji.

**Writing – original draft:** Thinley Dorji.

**Writing – review & editing:** Dorji Penjor, Thinley Dorji.

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
