## [Decision Letter · Decision Letter 0]

15 Jul 2020

PONE-D-20-09133

Circumstances of human conflicts with bears and patterns of bear maul injuries in Bhutan: review of records 2015-2019

PLOS ONE

Dear Dr. Dorji

Thank you for submitting your manuscript to PLOS ONE. After careful consideration, we feel that it has merit but does not fully meet PLOS ONE’s publication criteria as it currently stands. Therefore, we invite you to submit a revised version of the manuscript that addresses the points raised during the review process.

We look forward to receiving your revised manuscript.

Kind regards,

Tunira Bhadauria, Ph.D.

Academic Editor

PLOS ONE

Journal Requirements:

2. In the ethics statement in the manuscript and in the online submission form, please provide additional information about the patient records used in your retrospective study. Specifically, please ensure that you have discussed whether all data were fully anonymized before you accessed them and/or whether the IRB or ethics committee waived the requirement for informed consent. If patients provided informed written consent to have data from their medical records used in research, please include this information.

4. We note that Figure 1  in your submission contain map images which may be copyrighted. All PLOS content is published under the Creative Commons Attribution License (CC BY 4.0), which means that the manuscript, images, and Supporting Information files will be freely available online, and any third party is permitted to access, download, copy, distribute, and use these materials in any way, even commercially, with proper attribution. For these reasons, we cannot publish previously copyrighted maps or satellite images created using proprietary data, such as Google software (Google Maps, Street View, and Earth). For more information, see our copyright guidelines: http://journals.plos.org/plosone/s/licenses-and-copyright.

4.1.    You may seek permission from the original copyright holder of Figure 1  to publish the content specifically under the CC BY 4.0 license.

4.2.    If you are unable to obtain permission from the original copyright holder to publish these figures under the CC BY 4.0 license or if the copyright holder’s requirements are incompatible with the CC BY 4.0 license, please either i) remove the figure or ii) supply a replacement figure that complies with the CC BY 4.0 license. Please check copyright information on all replacement figures and update the figure caption with source information. If applicable, please specify in the figure caption text when a figure is similar but not identical to the original image and is therefore for illustrative purposes only.

5. Please upload a copy of Figure 3, to which you refer in your text on page 5. If the figure is no longer to be included as part of the submission please remove all reference to it within the text.

6. Please upload a copy of Supporting Information file which you refer to in your text on page 13.

Reviewers' comments:

Reviewer's Responses to Questions

**Comments to the Author**

1. Is the manuscript technically sound, and do the data support the conclusions?

Reviewer #1: Yes

Reviewer #2: Yes

2. Has the statistical analysis been performed appropriately and rigorously? 

Reviewer #1: N/A

Reviewer #2: Yes

3. Have the authors made all data underlying the findings in their manuscript fully available?

Reviewer #1: Yes

Reviewer #2: Yes

4. Is the manuscript presented in an intelligible fashion and written in standard English?

Reviewer #1: No

Reviewer #2: Yes

5. Review Comments to the Author

Reviewer #1: This article summarizes the incidence of personal injury as a result of bear attacks in Bhutan across a 5-year period. The descriptive statistics are compelling and should be available to the public. I highly recommend publishing the manuscript after it is revised. My primary concern is that the rationale for why this paper should be published was not substantive enough in the Introduction. In addition, given the seriousness of the issue and what appears to be a high incidence of bear-maul injuries, the conclusions were not strong enough and no recommendations were offered in the Discussion. What is the significance of this data?

I recommend that another round of copy-editing occur. Figure 1 is lovely, but the location of the specific hospital from which the data was collected (JDWNR) should be clearly indicated and in a bigger font size. In addition, it is not clear where the bears are most likely to interact with human settlements (paragraph 2 of the Introduction). The threat to conservation efforts was mentioned in the introduction and the discussion sections, but not explained sufficiently. The Results section dealt with the data from one hospital, but could the other two referral hospitals have also treated injuries due to bears? Is there any reason to account for the increase in incidents in 2017?

With revision, this paper will make a significant impact in the literature on bear attacks.

Reviewer #2: The authors should improve the conclusions section - presently, they do not suggest how human-bear conflicts can be reduced or how medical help to victims of bear attack can be made more accessible and improved. There is no suggestion on how conservation can be improved or what steps could be taken to reduce animal-human conflicts.

As the article is from Bhutan, and also gives a glimpse of the nature of injuries, the article should be accepted with the modifications suggested.

6. PLOS authors have the option to publish the peer review history of their article (what does this mean?). If published, this will include your full peer review and any attached files.

Reviewer #1: No

Reviewer #2: No

---

## [Author Response · Author response to Decision Letter 0]

30 Jul 2020

Reviewer 1 comments

Reviewer: Is the manuscript presented in an intelligible fashion and written in standard English? Reviewer #1: No, Reviewer #2: Yes

Response: We have revised the language and grammar throughout the manuscript.

Reviewer: This article summarizes the incidence of personal injury as a result of bear attacks in Bhutan across a 5-year period. The descriptive statistics are compelling and should be available to the public. I highly recommend publishing the manuscript after it is revised. 

Response: Thank you for providing insightful reviews. We have tried our best to revise our manuscript to reflect the valuable comments.

Reviewer: My primary concern is that the rationale for why this paper should be published was not substantive enough in the Introduction. 

Response: We have given the scenario of human-bear conflict in paragraph 3. This paper is intended to understanding the human cost of human-bear conflict while bear conservation efforts are equally important. We have indicated our reason for publishing this data as, “A better understanding of bear attacks may help in formulating specific measures to prevent human-bear conflict.”

Reviewer: In addition, given the seriousness of the issue and what appears to be a high incidence of bear-maul injuries, the conclusions were not strong enough and no recommendations were offered in the Discussion. What is the significance of this data?

Response: The two main significance are to document the grievous nature of injury and accessibility of surgical services to the victims and how human injuries are a threat to bear conservation efforts. We have revised the Discussion section to reflect these messages. We have revised the conclusion statement.

Reviewer: I recommend that another round of copy-editing occur. 

Response: We have revised the language and grammar throughout the manuscript.

Reviewer: Figure 1 is lovely, but the location of the specific hospital from which the data was collected (JDWNR) should be clearly indicated and in a bigger font size. 

Response: We have removed Figure 1 and presented the information on the number of victims by districts in the form of bar chart.

Reviewer: In addition, it is not clear where the bears are most likely to interact with human settlements (paragraph 2 of the Introduction). 

Response: In Bhutan, bears are most likely to interact with human settlements along the temperate zone at altitudes of 2000-4000 metres above sea level. (This has been indicated in paragraph 1 and beginning of paragraph 2).

Reviewer: The threat to conservation efforts was mentioned in the introduction and the discussion sections, but not explained sufficiently. 

Response: We have inserted a section on bear conservation efforts in the Discussion.

Reviewer: The Results section dealt with the data from one hospital, but could the other two referral hospitals have also treated injuries due to bears? 

Response: There are two regional referral hospitals in the country and the National Referral Hospital where the study was conducted. Till date, no ENT or plastic surgeons are posted in the two referral hospitals. Till now, the practice has been to refer the majority of the bear maul victims to the National Referral Hospital. From our clinical experience, only few cases might not have been captured in our study. We have reflected this limitation in the last paragraph of the Discussion section.

Reviewer: Is there any reason to account for the increase in incidents in 2017? With revision, this paper will make a significant impact in the literature on bear attacks.

Response: We do not postulate any reason on the increase in incidents in the year 2017. The incidence in absolute terms are within the capacity of the hospital to provide care for.

Reviewer 2 comments

Reviewer: The authors should improve the conclusions section - presently, they do not suggest how human-bear conflicts can be reduced or. 

… There is no suggestion on how conservation can be improved or what steps could be taken to reduce animal-human conflicts.

Response: We have inserted on section in the Discussion on how to reduce human-bear conflict in the context of Bhutan. We have written only briefly as the primary objective is to describe the human injuries and that the authors, who are medical doctors, may not be able to justify all the conservation efforts.

Reviewer: how medical help to victims of bear attack can be made more accessible and improved… As the article is from Bhutan, and also gives a glimpse of the nature of injuries, the article should be accepted with the modifications suggested.

Response: We have elaborated on the need to expand and increase the capacity of surgical services in the country, “While bear maul incidents were reported from 14 districts, access to surgical treatment is available only at three tertiary hospitals. Seven patients were referred outside the country for plastic and reconstructive surgery, the cost of which is borne by the Royal Government of Bhutan. This reflects the need to expand surgical services and increase the capacity of specialist surgical services in the country.”

---

## [Editor Report · Decision Letter 1]

4 Aug 2020

Circumstances of human conflicts with bears and patterns of bear maul injuries in Bhutan: review of records 2015-2019

PONE-D-20-09133R1

Dear Dr.Dorji

We’re pleased to inform you that your manuscript has been judged scientifically suitable for publication and will be formally accepted for publication once it meets all outstanding technical requirements.

Kind regards,

Tunira Bhadauria, Ph.D.

Academic Editor

PLOS ONE

Additional Editor Comments (optional):

I have gone through the revised manuscript and I am happy with the revision done by the authors , they have incorporated comments as well as suggestions from both the reviewers in the manuscript. Each comment has also been responded to separately by the authors in point wise manner, meeting the high standard of the journal and therefore making it acceptable for publication therein.
---

## [Editor Report · Acceptance letter]

10 Aug 2020

PONE-D-20-09133R1 

Circumstances of human conflicts with bears and patterns of bear maul injuries in Bhutan: review of records 2015-2019 

Dear Dr. Dorji:

I'm pleased to inform you that your manuscript has been deemed suitable for publication in PLOS ONE. Congratulations! Your manuscript is now with our production department. 

Kind regards, 

on behalf of

Dr. Tunira Bhadauria 

Academic Editor

PLOS ONE